# Addressing the role of centromere sites in activation of ParB proteins for partition complex assembly

Sylvain Audibert[1,☯,¤], Nicolas Tanguy-le-Gac[1,☯], Jérôme Rech[2], Catherine Turlan[2], Jean-Yves Bouet[2], Kerstin Bystricky[1,3], David Lane[2]*

1 Laboratoire de Biologie Moléculaire Eucaryote (LBME), Centre de Biologie Intégrative (CBI), CNRS, Université de Toulouse (UPS), Toulouse, France, 2 Laboratoire de Microbiologie et de Génétique Moléculaires (LMGM), CBI, CNRS, UPS, Toulouse, France, 3 Institut Universitaire de France (IUF), Paris, France

☯ These authors contributed equally to this work.
¤ Current address: Genome Damage and Stability, Brighton, England, United Kingdom
* dave@ibcg.biotoul.fr

**Data Availability Statement:** All relevant data are within the manuscript and its Supporting Information files.

## Abstract

The ParB-*parS* partition complexes that bacterial replicons use to ensure their faithful inheritance also find employment in visualization of DNA loci, as less intrusive alternatives to fluorescent repressor-operator systems. The ability of ParB molecules to interact via their N-terminal domains and to bind to non-specific DNA enables expansion of the initial complex to a size both functional in partition and, via fusion to fluorescent peptides, visible by light microscopy. We have investigated whether it is possible to dispense with the need to insert *parS* in the genomic locus of interest, by determining whether ParB fused to proteins that bind specifically to natural DNA sequences can still assemble visible complexes. In yeast cells, coproduction of fusions of ParB to a fluorescent peptide and to a TALE protein targeting an endogenous sequence did not yield visible foci; nor did any of several variants of these components. In *E.coli*, coproduction of fusions of SopB (F plasmid ParB) to fluorescent peptide, and to dCas9 together with specific guide RNAs, likewise yielded no foci. The result of coproducing analogous fusions of SopB proteins with distinct binding specificities was also negative. Our observations imply that in order to assemble higher order partition complexes, ParB proteins need specific activation through binding to their cognate *parS* sites.

## Introduction

The hub of the mechanism that drives bacterial mitosis, or partition, is a complex formed by binding of a specific ParB protein to a small number of clustered *parS* binding sites. The *parS* array functions as a centromere, and the complex serves as a kinetochore by activating the corresponding ParA ATPase to segregate replicas of its own chromosome or plasmid to incipient daughter cells. ParBs of most low copy-number plasmids and of all known chromosomes bind

**Funding:** JYB; IBM; Agence Nationale de la Recherche KB; SPAREDAM, ANDY; Agence Nationale de la Recherche https://anr.fr/ The funders had no role in study design, data collection and analysis, decision to publish, or preparation of the manuscript.

to their cognate *parS* sites as dimers, primarily via a helix-turn-helix (HTH) motif. They also bind, more weakly, to non-specific DNA. But unlike other proteins that bind through HTH motifs, notably transcription regulators, ParB proteins self-associate (oligomerize) and so pervade nearby DNA to enlarge their complex, a process termed "spreading". Spreading is not only integral to the partition process but has also enabled use of ParB/S systems as an alternative to fluorescent repressor-operator systems (FROS) for visualization of genetic loci in bacteria [1,2]. We have developed them for use in eucaryote cells and viruses as the ANCHOR system [3–5]. They offer certain advantages over FROS: the weakness of ParB oligomerization and DNA binding interactions allows other chromatin-based processes to disperse the complexes easily, making them less disruptive than FROS, and the small number of integrated *parS* binding sites involved is less locally intrusive than the hundreds typical of FROS. Nevertheless, dispensing with the need for prior *parS* integration through direct binding to endogenous sequences would eliminate potential artifacts of even such minor genome modification and would greatly streamline the procedure. Fusion of the ParB and fluorescent peptide (FP) components to proteins whose binding can be tailored to recognize natural genome sequences— TALE and Cas9—might allow specific tagging of unaltered sites while preserving the advantages of ParB/S systems. However, this would work only if ParBs can spread without first binding to their *parS* sites. It was not clear that they can. To assess the feasibility of removing the *parS* integration step from the ANCHOR system we have aimed in the work reported here to identify the interactions that enable ParB spreading.

HTH-type ParB proteins share a broadly consistent three-domain organization of both structure and function. The intrinsically disordered N-terminal domain interacts with ParA [6] and with itself [7], the central domain contains the motifs responsible for specific *parS* recognition and for DNA binding [8–12], and the C-terminal domain dimerizes the protein [6,7,9,13] to form the basic active ParB unit; in certain ParBs the latter also plays roles in DNA binding and ParA interaction [14–16].

Early studies of mutant ParBs of the P1 and F plasmids implicated the N-terminal domain of these proteins in spreading [8,17–19], as also confirmed later for chromosomal ParBs [14,20], and suggested that spreading is needed for partition. These observations, together with the demonstration that the N-terminal domain oligomerizes P1 ParB dimers [7], the known binding of HTH proteins to non-specific DNA and the small number of foci seen in cells containing fluorescent ParB derivatives [21,22] led to the general view that ParB bound in the core complex recruits further ParB molecules whose weak interactions with themselves and with neighbouring non-specific DNA create a metastable complex large and cohesive enough to activate partition [23]. Spreading was initially envisaged as proceeding laterally from the centromere along adjacent DNA. However, certain observations were inconsistent with this view [24,25], and Bouet *et al.* [26] proposed that ParB spreads not only *in cis* from the nucleating complex but also *in trans* to the nucleoid and to distant sites on the same molecule, like bees round a hive rather than birds on a wire. The transient bridging (*trans*) and looping (*cis*) interactions and the indeterminate form of the complex implied by this proposal have since been substantiated and refined by studies of complexes formed by the ParB proteins of several species [27–31], and the idea has recently been extended to the chromatin realm [32].

However, the full role of centromere binding in formation of higher-order partition complexes is not yet understood. Since spreading is not seen to occur spontaneously, in the absence of *parS*, it would appear to need a specific switch in ParB conformation. Is this induced directly upon binding to *parS*? Or is it a consequence of the oligomerization interaction of ParB N-terminal domains, with *parS* binding serving only to focus and anchor the complex? (Fig 1) On one hand there are indications that the properties of ParBs do change in response to centromere binding: F SopB and P1 ParB co-repressor activity is stimulated *in trans* by *sopC* and

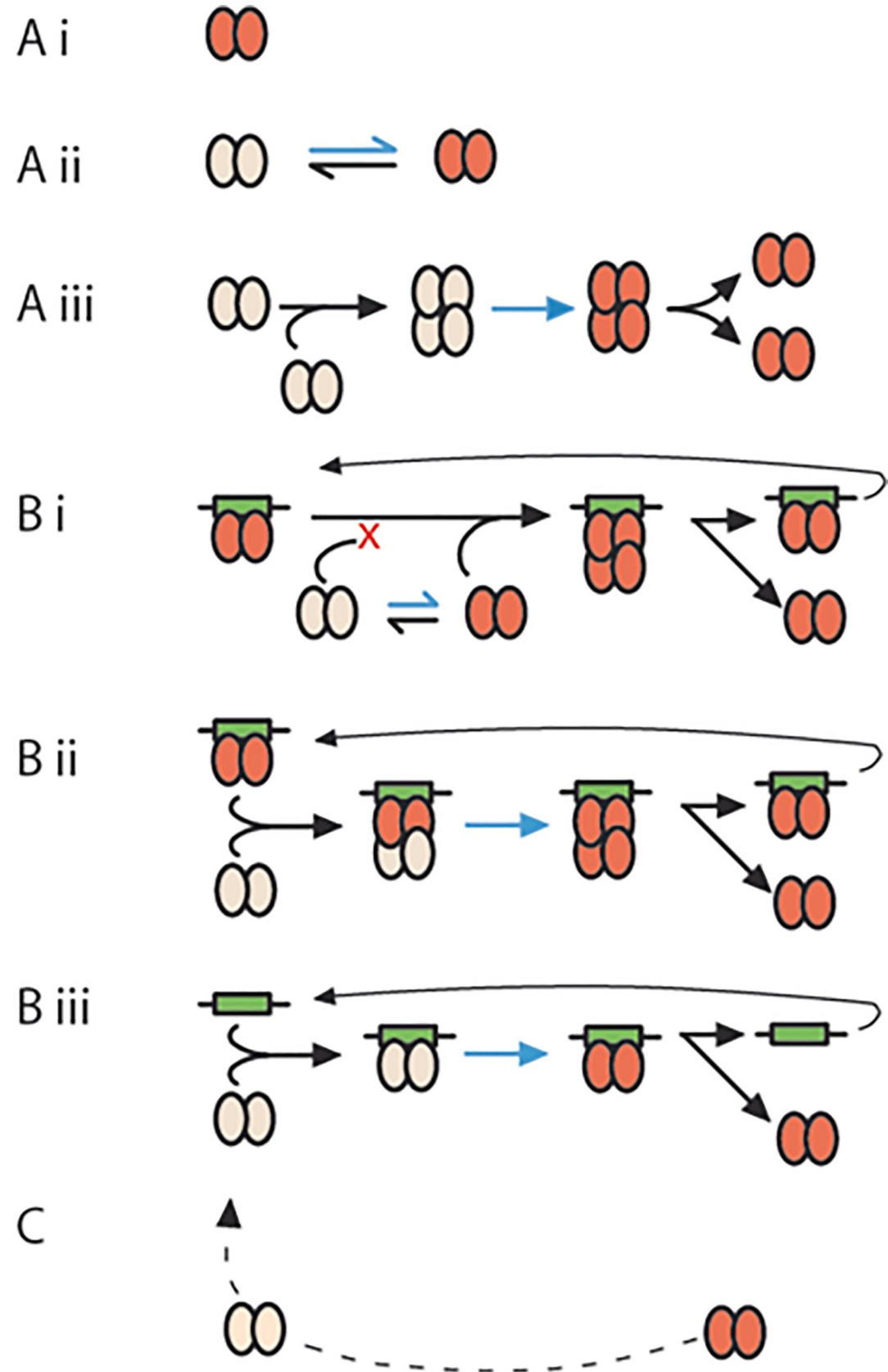

**Fig 1. ParB activation mechanisms.** Possible mechanisms responsible for a conformation change in ParB dimers that enables them to enlarge the partition complex through oligomerization, i.e. to spread. Several, non-exclusive, activation mechanisms can be envisaged, it being assumed that the active conformation is maintained for a significant fraction of the partition complex lifetime. Dimers able to spread are shown in red, those unable to in grey. The activating conformational change is depicted by blue arrows. **A.** mechanisms independent of the centromere: **(i)** ParB dimers are intrinsically capable of oligmerization and need no specific activation. **(ii)** Two forms of ParB, able and unable to oligomerize, are interconvertible via a spontaneous conformational shift. **(iii)** An initial dimer-dimer interaction induces the activation that allows the dimers to then participate independently in spreading. **B.** mechanisms requiring the centromere: **(i)** Centromere-bound dimers activated by binding or withdrawn from a pool in conformer equilibrium, as in Aii, interacts selectively with other active dimers, thus trapping them near the centromere and shifting the diffusible ParB equilibrium towards the active form. **(ii)** Inactive dimers are activated through contact with already active dimers residing on the centromere (or with previously released activated dimers). **(iii)** Successive binding of inactive dimers, activation by the centromere and release builds a pool of active dimers. **C.** Loss of the conformation enabling oligomerization could return dimers to the inactive pool for recycling.

*parS* respectively [33,34], SopB-mediated stimulation of SopA ATP hydrolysis is enhanced by *sopC* [35] and a newly-discovered CTPase activity exhibited by some ParB proteins, including SopB, is enhanced by centromere binding [36,37]. On the other hand, Surtees & Funnell [7] observed oligomerization of P1 ParB N-terminal domains in the absence of *parS in vitro* and in yeast, and Hyde *et al*. [38] concluded that specific binding of KorB did not reduce intrinsic disorder but rather selected from a population of natural conformers.

We report here our attempts to observe spreading of ParBs fused to non-*parS* DNA-binding proteins, and to distinguish ParB-*parS* binding from ParB-ParB interaction as the basis for conversion to spreading competence.

## Materials and methods

### Strains

**Bacteria.** *E.coli* K12 strains used in microscopy and gene expression experiments were derivatives of W1485, as detailed in Table 1 and schematized in S1 Fig. Transformation recipients were DH5α and DH10B [39] except for constructions involving recombination-prone RVD repeats of *tal* genes where SURE2 (Stratagene) or a derivative cured of the F', D111, was

**Table 1. Bacterial strains.**

| Strain | Genotype/relevant properties | Source |
|---|---|---|
| CB0129 | *thi*, *leu*, *thyA*, *deoB*, *supE* | [42] |
| DLT812 | CB0129 Δ(*ara-leu*)7696, *zac3051*::Tn*10* | [43] |
| DLT1215 | DLT812 *rpsL812* | [44] |
| DLT1912 | DLT812 ΩP*cp*18::*araE*533, by P1-transduction with *frt.kan.frt* and Flp-excision of *frt-kan* | this work |
| DLT2067 | MC1061 ΩP*cp*18::*araE*533, Δ*ara*(*FGH*), λRS45-*sopC-rpsL*+-*cat* | [44] |
| DLT2074 | DLT1215 *xylE*::*sopC* | [31] |
| D135 | DLT1912 Δ*ara*(*FGH*), by P1 transduction with *frt.cat.frt* and Flp-excision of *frt.cat* | this work |
| D143 | D135 λRS88-*frt.aadA.frt*-4xIR-p*cry*::*lacZ* lysogen, transfer from pDAG545 | this work |
| D150 | D135 λRS45-*sopC-rpsL*+-*cat* lysogen, transduction from DLT2067 | this work |
| D179 | D135 Δ*lacZ4787*, Δ*lacA*, by transduction with *frt.kan.frt* from JW0333 (Keio collection) and Flp-excision of *frt.kan* | this work |
| D183 | D143 Δ*lacZ4787*, Δ*lacA* by transduction with *frt.kan.frt* from JW0333 (Keio collection) and Flp-excision of *frt.kan*, *frt.aadA* | this work |
| D195 | D179 λRS88-*sopC-pldc* lysogen, transfer from pDAG418 | this work |

used. The host for recombinational transfer of centromere sequences from plasmids to λ phages [40] was MC1061 [41].

**Yeast.**    The progenitor of the *Saccharomyces cerevisiae* strains (Table 2) was strain W303, a gift from Frederic Beckouet. It was sequentially modified by (i) Cas9-mediated insertion of a Gal:HO cassette at ade3—W303 cells were transformed by ade3::Gal:HO DNA with the plasmids p414-TEF1p-Cas9-CYC1t and pRPR1-gRNA.ADE3-RPR1t, (ii) deletion of the *ho* gene by insertion and excision of a URA3 cassette, and (iii) lithium acetate transformation with a cassette of URA3 flanked by sequences at Chr3:197kb, yielding ySA46. ySA27 was obtained by substituting the URA3 cassette of ySA46 for a ANCH3 cassette, with 5-fluoroorotic acid selection for ura⁻.

## Plasmid constructions

Construction outlines are given here; details are available on request. Relevant characteristics of plasmids are given in Table 3.

**ANCHOR system visualization.**    Tale constructs were based on the pZHY501 shuttle (*S. cerevisiae-E.coli*) vector, provided by Daniel Voytas (via Addgene) [45], which carries the Nt and Ct (non-RVD) domains of the *X.oryzae* Tale PthXo1 gene fused to the FokI nuclease coding sequence. pZHY501was modified by site-specific mutagenesis to introduce sites for *Avr*II and *Nru*I immediately upstream of the Tale *Nt* sequence, and by deletion of the *fokI* gene using *Bam*HI, *Bsa*BI and Klenow polymerase, yielding pVR203. Repeat variable di-residue (RVD) domains specific for URA3 Nter nt 17–32 (U3aL) and Cter nt 632–648 (U3bR) were obtained as *Bsm*BI site-ended PCR products from plasmids kindly provided by Bing Yang [46], and inserted between the *Bsm*BI sites in pVR203 to create the *Nt::ura3::Ct* fusions in pVR204 (U3aL) and pVR206 (U3bR). A codon-optimized synthetic *or3* (*parB*) gene [4] was obtained as a PCR product with terminal *Avr*II and *Nru*I sites and inserted between these sites in pVR204 to create the *or3s::tal.U3aL* fusion in pSA316. The stop>leu-mutated codon of the same *or3s* gene was fused to codon 2 of the mCherry coding sequence; the *or3s::mCh* fusion was inserted into a vector then excised with *Kpn*I and *Not*I and inserted between these sites of pRS424 [47], yielding pSA312.

**SopB::mVenus visualization.**    Plasmids producing SopB::mVenus were pCAT10, made by joining *ori*$^{\text{PSC101}}$-p*lac*, *cat* and *sopB*$^{\text{R219A}}$::*mVenus* PCR fragments, and pJYB294, the *sopB*⁺, *aadA* equivalent. Plasmids producing SopB::dCas9 were derived from the p15A-based plasmid pdCas9 ([48]; Addgene #44249): substitution of *aadA* (Sp$^{\text{R}}$) for *cat* gave pCAT02, insertion of *sopB*$^{\text{R219A}}$ and linker (ELGSG)-dCas9 5'-terminus PCR fragments into pCAT02 gave pCAT05, and an equivalent insertion of *sopB*⁺ into pdCas9 gave pCAT15. Plasmids producing sgRNAs were derived from pgRNA (Addgene #44251): deletion of the promoter-gRNA interval by inverse PCR gave pCAT06, while replacement of this interval by a 20bp sequence from *sopC* (one arm of the palindrome and its flank, to avoid hairpin formation), *xylE* and an unrelated MS2 sequence ("random") gave pCAT08, -184 and -168 respectively. *AadA* was substituted for *cat* in pwtCas9 (Addgene #44250) to give pCAT13.

**Table 2. Yeast strains.**

| Strain | Genotype/relevant properties | Source |
|---|---|---|
| W303 | leu2-3,112 trp1-1 can1-100 ura3-1ade2-1 his3-11,15 | F. Beckouet |
| ySA27 | W303 Δho ade3::Gal-HO Anch3@chIII.197kb | this work |
| ySA46 | W303 Δho ade3::Gal-HO URA3@chIII.197kb | this work |

**Table 3. Plasmids.**

| Name | Relevant characteristics | Source (this work, except as noted) |
|------|--------------------------|-------------------------------------|
| pGB2 | *ori-rep*$_{pSC}$, *aadA* | [50] |
| pDAG123 | *ori*$_{pBR}$, p*ldc-lacZYA*, *kan*, *bla* | [43] |
| pDAG170 | *ori*$_{pBR}$, *araC-paraBAD-sopA'sopB*$_F$, *cat* | [26] |
| pNR120 | *ori-rep*$_{pSC}$*, p*LtetO-sopA*$^{N15}$-*sopB*$^{N15/F}$, *cat* | [9] |
| pRS415 | *ori*$_{pBR}$, *bla*, *lacZYA*; promoter assay vector | [40] |
| pZS*21 | *ori-rep*$_{pSC}$*, p*LtetO*, *kan* | [51] |
| pCAT02 | *ori*$_{p15A}$, p*otetA -dCas9*, *aadA* | |
| pCAT05 | *ori*$_{p15A}$, p*otetA -sopB*$^{R219A}$::*dCas9*, *cat* | |
| pCAT06 | *ori*$_{pUC}$, p*J23119-sgRNA*$^0$, *bla* | |
| pCAT08 | *ori*$_{pUC}$, p*J23119-sgRNA*$^{sopC}$, *bla* | |
| pCAT10 | *ori*$_{pSC}$, p*lac-sopB*$^{R219A}$::mVenus, *aadA* | |
| pCAT13 | *ori*$_{p15A}$, p*otetA -Cas9*, *aadA* | |
| pCAT15 | *ori*$_{p15A}$, p*otetA -sopB*$^+$::*dCas9*, *cat* | |
| pCAT168 | *ori*$_{pUC}$, p*J23119-sgRNA*$^{MS2}$, *bla* | |
| pCAT180 | *ori*$_{pUC}$, p*J23119-sgRNA*$^{sopC2}$, *bla* | |
| pCAT184 | *ori*$_{pUC}$, p*J23119-sgRNA*$^{xylE}$, *bla* | |
| pJYB294 | *ori*$_{pSC}$, p*lac-sopB*$^+$::mVenus, *aadA* | [31] |
| pSA312 | *ori*$_{2\mu}$, *ori*$_{pBR}$, p*Cyc1-or3*::*mCh*, *trp1* | |
| pSA316 | *cen6-ars4*, *ori*$_{pBR}$, p*Tefα-or3*::*tal.U3aL*, *leu2* | |
| pNR129 | *ori-rep*$_{pSC}$*, p*LtetO-sopA*$^{N15}$Δ(cdns 4–248)-*sopB*$^{N15/F}$::*megfp*, *kan* | |
| pNR195 | miniP1 (*ori-repA-parABS*), p*ara*, *cat* | |
| pNR197 | miniP1 (*ori-repA-parABS*), p*ara-sopA*$^{N15}$Δ(4–248)-*sopB*$^{N15}$::*megfp*, *cat* | |
| pNR189 | miniP1 (*ori-repA-parABS*), p*ara-sopA*$^{N15}$Δ(4–248)-*sopB*$_{N15}$, *cat* | |
| pNR198 | miniP1 (*ori-repA-parABS*), p*LtetO-sopA*$^{N15}$Δ(40–355)-*sopB*$_{F.R219A}$::*megfp*, *bla* | |
| pDAG418 | *ori*$_{pBR}$, *sopC* | |
| pDAG525 | miniP1 (*ori-repA-parABS*), p*LtetO-sopA*$^{N15}$Δ(40–355)-*sopB*$_F$::*megfp*, *cat* | |
| pDAG541 | pRS415 4xIR-p*cry*::*lacZ* | |
| pDAG545 | pDAG541 *frt-aadA-frt* | |
| pDAG607 | *ori-rep*$_{pSC}$, p*ara-sopA*$^{N15}$Δ(40–355)-*sopB*$_F$, *aadA* | rep$_{pSC}$* is a low copy-number mutant of pSC101 isolated by Xia *et al.* [52] |

## SopB::mEgfp visualization

Plasmids carrying the hybrid *sopB* (N15 codons 1–175: F codons 174–323) transcribed from the p*L-tetO* promoter were derived from pNR120 by in-frame deletion of *sopA* residues 4–248 (pNR123), replacement of *cat* by *kan* (pNR127) and fusion of *sopB*$^{N15/F}$ 5' to *megfp*, yielding the signalling plasmid pNR129.

Plasmids carrying *sopB*$^{N15}$ were derived from pZC326 [44], a mini-F—mini-P1 hybrid into which we inserted a high copy-number origin, making pDAG382, to facilitate construction. To place *sopB*$^{N15}$ under *araC-paraBAD* control we substituted it for *sopB*$^F$ in pDAG170 [26]. The *araC-para-sopA'-sopB*$^{N15}$ expression unit was joined to the mini-P1 portion of pZC326 to make the binding plasmid pNR189, and *sopA'-sopB*$^{N15}$ deleted from pNR189 to make the control plasmid pNR195. The *megf* gene (see below) was amplified by PCR and fused to the *sopB*$^{N15}$ 3' end in pNR189 to make pNR197.

**Table 4. Oligonucleotides for N15 "centromere".**

| | |
|---|---|
| f1t | 5'-AATTCTTCTTCCGGCT**GTGCGACCAC**GGTCGCAC**C**ATTCCGTTGG |
| f1b | GAAGAAGGCCGA**CACGCTGGTG**CCAGCGTG**G**GTAAGGCAACCACGT-'5  IR1 |
| f2t | TGCAGTCAAAGAGG**GTGCGACCTC**GGTCGCAC**G**AGATAATGAA |
| f2b | CAGTTTCTCC**CACGCTGGAG**CCAGCGTG**G**GAGATAATGAATCGA-'5  IR2 |
| f3t | AGCTGTCTGATATC**GTGCGACCAT**GGTCGCAC**G**GAATAGAAAT |
| f3b | CAGACTATAG**CACGCTGGTA**CCAGCGTG**C**CTTATCTTTACATG-'5  IR3 |
| f4t | GTACGTCCGCTTTC**GTGCGACCAC**GGTCGCAC**G**CTTTTCCATTCT |
| f4b | CAGGCGAAAG**CACGCTGGTG**CCAGCGTG**C**GAAAAGGTAAGACTAG-'5  IR4 |

Plasmids carrying *sopB*[F] were constructed as follows. pDAG607: transfer of the *araC-para-sopA'-sopB*[F] unit from pDAG170 to pGB2. pDAG525: fusion of *sopA'-sopB*[F] to *egfp* downstream of p*LtetO* in a pZS21* vector, followed by successive replacements of *egfp* by *megfp*, *kan* by *cat*, and the vector by the mini-P1 segment of pZC326. pNR198: replacement of *sopB*[+] in pDAG433 [9] by *sopB*[F.R219A] amplified from pJYB223 [12], in-phase insertion of *megfp* 3' to the *sopB*[F.R219A], and joining of the p*LtetO-sopA'-sopB*[F.R219A]::*megfp-bla* segment to the mini-P1 portion of pZC326.

Plasmids for centromere transfer to attλ: the *sopC* sequence was inserted between the *kan* gene and the p*ldc* promoter in pDAG123 [43] to give pDAG418. A *frt-aadA-frt* cassette was inserted upstream of the 4xIR unit (see below) in pDAG541 to give pDAG545.

**4xIR construction.** To make an N15 centromere comparable to *sopC* of F, we joined the four IR sites to each other such that the centres of adjacent IRs are separated by 43bp. Four pairs of complementary oligonucleotides, corresponding to each of the natural IRs and its flanks (Table 4), were designed to form duplexes with 5' extensions permitting ligation in a defined order. All oligonucleotides except f1t and f4b were phosphorylated with T4 polynucleotide kinase, annealed pairwise by heating and slow cooling, then mixed and incubated with T4 DNA ligase. Ligation products were amplified by PCR using f1t and f4b as primers, and the PCR products digested with *Apa*LI, *Hin*dIII and *Bsr*GI to remove self-ligated products, size-selected by gel electrophoresis, phosphorylated, and ligated with *Sma*I-digested, dephosphorylated pRS415, yielding pDAG541.

**Monomer Egfp mutant.** To impede the tendency of the original Egfp (Clontech) to dimerize we introduced the A206K mutation by strand-overlap extension PCR [49] using mutagenic primers A206K.H3-3 and A206K.H3-4 and flanking primers SopBN15.949–972 and Xba-dsegfp. The *egfp*[A206K] product was initially fused to *sopB*[N15] (pSA579), then to *sopB*[F] with an *Xho*I site between the *sopB* and *egfp* genes (pDAG524), and finally to *sopB*[N15/F] (pNR129). Primers (mutagenic bases underlined):

| | |
|---|---|
| A206K.H3-3 | CCCAGTCC<u>A</u>AGCT<u>T</u>AGCAAAGACCCCAACG |
| A206K.H3-4 | CTTTGCT<u>A</u>AGC<u>TT</u>GGACTGGGTGCTCAGGTAG |
| SopBN15.949–972 | GCAGAATTAGGTGCAGCTGAGCAG |
| Xba-dsegfp | GAATTCTAGAGTCGCGGCCGCTTTACTTG |

## Media and growth conditions

**Bacteria.** Routine cultures were grown with aeration at 37°C in Luria-Bertani broth supplemented as appropriate with (µg/mL) kanamycin (15), chloramphenicol (10), and

spectinomycin (20), or at twice these concentrations for solid (1.5% agar) media, and with ampicillin (100) and tetracycline (10) for both.

Cultures for gene expression and microscopy were grown with aeration at 30˚C in M9 salts supplemented with thymine (20μg/mL), Casamino acids (0.2%), thiamine (1μg/mL), 0.2% glucose or glycerol (MGC and MglyC respectively), antibiotics as above and inducers IPTG, anhydrotetracycline (aTc) and arabinose as needed (see legends).

**Yeast.** The basic medium was SC: 0.67% yeast nitrogen base (Difco) supplemented with all amino-acids except those used for selection, uracil, adenine and 2% glucose.Strains W303, ySA27 and ySA46 transformed with pSA312 alone or with pSA312 and pSA316 were grown overnight in SC lacking leucine (SC-LEU) or leucine and tryptophan (SC-LEU-TRP) respectively.

**DNA manipulation.** *In vitro* manipulation and construction of plasmids employed standard materials and procedures. DNA polymerases in PCR were Phusion (New England Biolabs) for synthesis of DNA used in constructions and DreamTaq (Thermo Fisher) for routine screening.

**SopB::Cas9 killing assay.** Binding of dCas9 to the site targeted by the sgRNAs used in this study was verified by testing the ability of wtCas9 at low concentration to cleave target DNA. The test strain DTL2074 was co-transformed by the compatible plasmid pairs to be tested, one expressing the sgRNA, the other wtCas9 (pCAT2) or dCas9 (pCAT13). Transformed bacteria were grown without anhydrotetracycline on plates selective for both plasmids (with Sp and Ap) for 24 hours at 37˚C. Specific Cas9 binding to the targeted site was determined by the absence of colony growth in the presence of wtCas9 but not of dCas9.

## Microscopy

**Bacteria.** Colonies of strains freshly transformed with plasmids carrying the *sopB* genes to be tested were used to inoculate MGC or MglyC at a concentration permitting at least 10 generations of exponential growth (doubling times of ~60 and ~120 minutes respectively), and incubated at 30˚C to an optical density at 600nm of 0.1–0.2 for viewing mEgfp fluorescence or 0.3–0.4 for mVenus. Samples were applied as 0.7μL drops to the surface of a layer of 1% agarose in growth medium, as described [53]. The cells were viewed at 30˚C using an Eclipse TI-E/B wide field epifluorescence microscope with a phase contrast objective (CFI Plan APO LBDA 100X oil NA1.45) and a Semrock filter YFP (Ex: 500BP24; DM: 520; Em: 542BP27) or FITC (Ex: 482BP35; DM: 506; Em: 536BP40). Images were taken using an Andor Neo SCC-02124 camera with illumination at 80% from a SpectraX source Led (Lumencor) and exposure times of 0.5-1second. Nis-Elements AR software (Nikon) was used for image capture and editing.

**Yeast.** Live-cell microscopy was performed as described [54], using an Olympus IX-81 wide-field fluorescence microscope equipped with a CoolSNAPHQ camera (Roper Scientific) and a Polychrome V (Till Photon-ics) electric piezo accurate to 10 nm, and an Olympus oil immersion objective 100X PLANAPO NA1.4. Yeast cells were spread on a layer of SD-agarose (YNB + 2% dextrose + 2%(w/v) agarose) set in a microscope slide trough. mCherry signal was acquired in 3D as 21 focal planes at 0.2 μm intervals with an acquisition time of 300ms.

**Silencing assay.** Transcription reporter strains freshly transformed with plasmids carrying the *sopB* genes to be tested were grown as for microscopy (above) at 30˚C to an optical density of 0.1–0.2 and chilled on ice. Samples were removed for assay of β-galactosidase and measurement of optical density as described [55].

**Western blotting.** Cells from exponentially-growing bacterial cultures were resuspended in SDS-loading buffer (10% glycerol, 5% β-mercaptoethanol, 2.3% SDS, 62.5mM Tris-HCl pH

6.8, 0.25% bromophenol blue), heated at 95˚C for 5 minutes with occasional vortexing, and centrifuged. Samples corresponding to 0.033 $OD_{600}$ units were subjected to PAGE on 4–12% gradient gels run at 200V for 50 minutes in either MOPS or Tris-glycine buffer. Proteins were transferred semi-dry to PVDF (0.2μm) membranes using a Transblot Turbo apparatus (Biorad), then exposed successively, with intermediate washing, to Tween-milk blocking buffer, rabbit α-SopB (1:1000; Eurogenetec), goat anti-rabbit IgG coupled to Horse-radish per-oxidase (1:1000) and ECL reagent (Clarity Western) before scanning (Gel Doc). Exponential phase yeast cells were concentrated and broken with glass beads using a Beads Beater. After centrifugation to remove debris, protein concentrations of the supernatants were measured with Bradford reagent and 20μg of each heated in loading buffer and subjected to PAGE on Biorad 4–15% gradient gels in Tris-glycine. Transfer and immunodetection were as in A, except that anti-GFP was used.

## Results

We begin by describing two of our attempts to observe partition complex assembly primed by specific binding to DNA sites other than the ParB protein's own centromere. One employs a plasmid site inserted in the *E.coli xylE* gene, the other involves several sites, natural and exoge-nous, within *S.cerevisiae* chromosomes.

### *E.coli xylE*

The CRISPR-Cas9 system was used. A guide RNA with a 20 nt sequence of *xylE* (sgRNA-*xylE*) was co-produced with a polypeptide comprising the F plasmid SopB protein fused at its C-ter-minus to the enzymatically inactive dCas9 protein. The ability of this fusion to recognise its target was confirmed by the lethality of the SopB fusion to active Cas9 both in *xylE* [+] cells pro-ducing sgRNA-*xylE* and in cells with an insertion of the F *sopC* centromere producing sgRNA-*sopC* (S1 Table). Western blot analysis confirmed production of adequate quantities of the dCas9 fusion proteins; despite variability in immunostaining signal, we could estimate that the SopB-fusion proteins were present at a minimum of 300 monomers per cell (S2A Fig). Ability of *xylE*-bound SopB::dCas9 to prime spreading was tested by observing formation of fluorescent foci in cells also producing SopB::mVenus, a fusion protein known to act normally in complex assembly and plasmid partition [53]. To prevent saturation of the incipient parti-tion complex by SopB::dCas9, its production was kept to a minimal level by allowing transcrip-tion from p*otetA* only at the basal, uninduced level, while strongly inducing production of SopB::mVenus. Compact foci of normal number and distribution appeared in *xylE*::*sopC* cells (Fig 2A); this was due to direct binding of SopB::mVenus *to sopC*, unmediated by the dCas9 fusion, since the foci also formed when the guide RNA carried a random sequence (Fig 2C). Cells without *sopC* expressing the same sgRNA and fusion genes showed no foci, only evenly distributed fluorescence (Fig 2B and 2D).

It is possible that the single *xylE* binding site used does not form a core complex sufficiently robust to trigger spreading, even though a single SopB binding site does so (S3 Fig). So we redid the experiment with a tandem-repeat binding sequence that resembles the natural *sopC* centromere, using a guide RNA sequence corresponding to eight of the ten functional 43bp repeats that make up *sopC* [56]. The R219A mutant derivative of SopB was used in the fusion proteins to prevent specific binding to *sopC* while still allowing the non-specific DNA binding needed for spreading [12]. Fig 2E and 2F shows that this modification did not enable focus for-mation either.

### *S.cerevisiae URA3*

Several ParB proteins, when fused to fluorescent peptides, form visible complexes in yeast strains engineered to harbour small arrays of their *parS* binding sites [3]. We examined the ability of such fusion proteins to form foci in the absence of their cognate *parS* sites when specific binding was provided by TALE proteins [57]. Fig 3 shows the results of a representative experiment, employing an experimental format similar to that of Fig 2. Yeast cells transformed with a plasmid from which the ParB fusion protein Or3::mCherry is produced formed one distinct focus in each nucleus, as expected for cells in G1, provided they have an integrated copy of Or3's cognate centromere site, Anch3 (Fig 3A). No foci were seen in cells of the parental strain, which lacks this site (Fig 3B and 3C). When Or3::mCherry was coproduced with a fusion of Or3 to a Tale peptide known to bind specifically to the 5' end of the *URA3* gene (Or3::Tale.U3L), it still formed foci in the Anch3 strain (Fig 3D) but also still failed to in cells without Anch3, whether or not they contained the Tale.U3L binding site (Fig 3E and 3F).

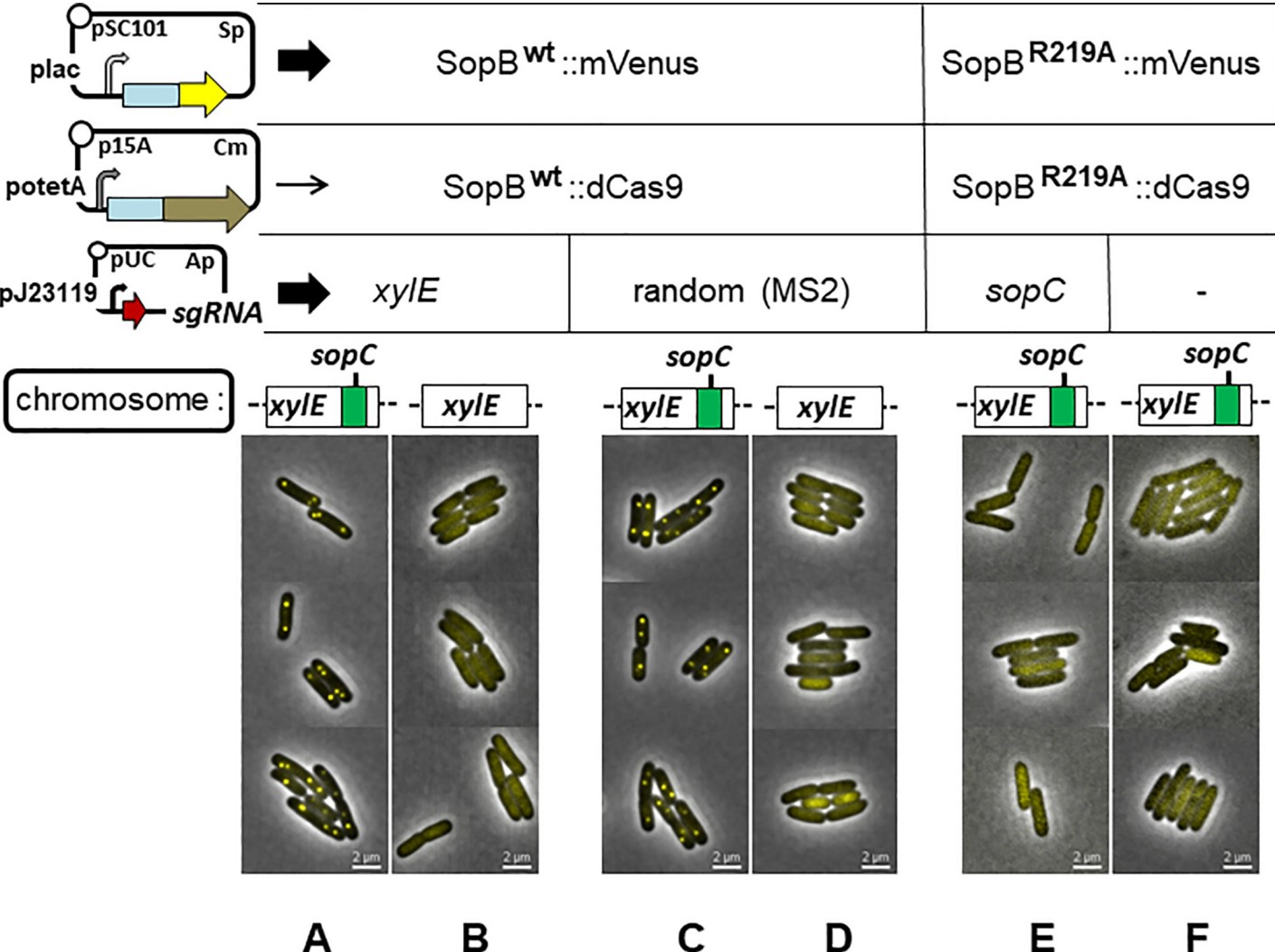

**Fig 2. Test of SopB spreading from dCas9-DNA complexes.** Cells of strain DLT1215 and its *xylE*Ω*sopC* derivative, DLT2074, carrying the plasmids that enable SopB visualization and SopB::dCas9 binding (upper left) were applied to buffered agarose-coated slides after at least 10 generations of exponential growth in MGC medium supplemented with and 30μM IPTG; arrow width next to each plasmid indicates the relative level of RNA produced from each. Each column of images shows cells grown with the combination of dCas9 or SopB target sequence, sgRNA, SopB::dCas9 and SopB::mVenus fusion directly above.

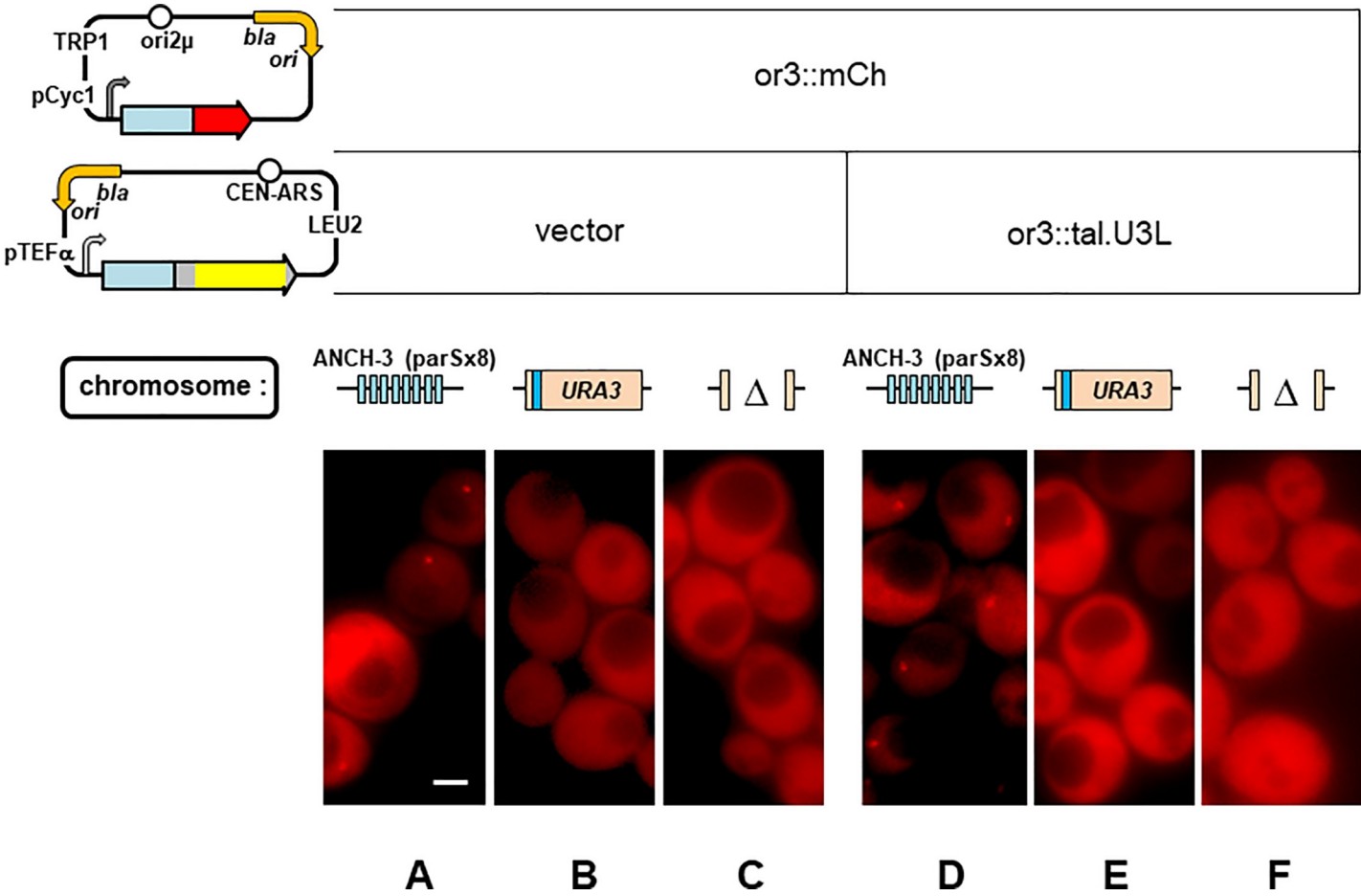

**Fig 3. Test of spreading by ParB specifically bound via fusion to Tale proteins in yeast.** Or3::mCherry fusion protein production from pSA312 in: **A.** strain ySA27, with Anch3 inserted, **B.** ySA46, with no insertion, **C.** W303, with a deletion in URA3 removing the specific Tal.U3L binding site (blue strip). Or3::mCherry production together with Or3::tal.U3L from pSA316 in strains ySA27 (**D**), ySA46 (**E**) and W303 (**F**). Fusion protein elements are or3 (blue), mCherry (red), tale RVDs (yellow) and backbone (grey).

This outcome, no foci at all, or the other, formation of aggregates rather than *bona fide* foci, was the result of all variations tried: several different ParBs, use of tripartite fusions (ParB::FP::Tale), inverting the gene order of fusions, replacing ParBs by their Nter domains, use of stronger and weaker promoters, targeting different specific sites, use of dCas9 fusions. This was in spite of confirmation that the full-sized fusion proteins were present and constituted the bulk of the plentiful fluorescence in the nuclei (S2B Fig).

Our inability to see fluorescent foci representing higher-order partition complexes with either the bacterial or the yeast test systems is most simply explained as the requirement for *parS* binding of at least some of its cognate ParB molecules to enable spreading. If so, this requirement leads to the question of whether all ParB dimers must contact their *parS* for this to happen, or whether much of the postulated conformation change can be effected at the protein level.

### Spreading of hybrid SopB proteins

It should be possible to distinguish ParB-*parS* binding from ParB dimer-dimer interaction as the event that enables spreading by using hybrid proteins with centromere-binding and N-

terminal domains of distinct specificities. A minimal complex seeded with limiting amounts of one ParB (the binding protein) might be expanded to a large complex upon provision of a second, hybrid ParB (the signalling protein) that shares the N-terminal domain but does not bind to the same *parS*: expansion of the complex could be observed by tagging the second ParB with a fluorescent peptide, and would indicate that direct binding to *parS* is not needed for spreading. The experimental set-up is similar to that of Fig 2, but here the initiating complex is natural and known to trigger spreading, and no bulky, potentially interfering foreign protein is involved.

The closely-related Sop partition systems of plasmid F and prophage-plasmid N15 appeared suitable for applying this approach. Bacteriophage N15 has sequence, structural and physiological similarities to lambdoid phages [58], but unlike the integrated λ prophage, N15 prophage is a linear, low copy number plasmid whose stable inheritance requires active partition. The SopB proteins of F and N15 are very similar, at 49% amino acid identity; SopB of F functions only with its cognate binding site (10 tandem copies in the F centromere, *sopC* [56]), not with those of N15 (IR1-4; [59]) [9]; and many N15:F hybrid proteins are functional, interacting with their SopA and centromere partners with the expected specificity [9]. One of these SopB proteins, SopB$^{N15/F}$ (termed hybrid 10 by Ravin *et al.*, [9]), comprises the N-terminal domain of N15 SopB and the DNA-binding and dimerization domains of F SopB (Fig 4, top left). It should be able to interact via its N-terminal domain with N15 SopB bound to IR centromere sites, but be unable to bind to these sites itself. The distribution of fluorescent SopB$^{N15/F}$ protein confirms this specificity: discrete foci are seen in cells with *sopC* integrated as part of a prophage at attλ (Fig 4A), whereas in cells with an analogous N15 centromere-site array (4xIR), in which N15 SopB forms normal foci (Fig 4C), SopB$^{N15/F}$::megfp fluorescence diffuses evenly throughout the cell (Fig 4B). The SopB$^{N15/F}$::megfp fusion is thus sufficiently specific to serve as a signalling protein.

To ensure that the binding proteins did not saturate spreading capacity, we assayed SopB-mediated silencing to estimate appropriate production levels. The prophage vector of the 4xIR array also carries a weak cryptic promoter, about 50bp further downstream, from which *lacZ* is transcribed, providing a sensitive measure of expansion of the partition complex in response to induced SopB$^{N15}$ synthesis. Silencing by the SopB$^{N15}$ protein and its mEgfp fusion derivative respond similarly to arabinose-mediated induction (S4A Fig), becoming discernible between 0.1 and 0.3 μM arabinose and strong above 0.6μM. This result is mirrored by focus formation, monitored in parallel (Fig 4C)—SopB$^{N15}$::megfp foci are scarcely seen at 0.1μM, present in most cells and discrete, though small, at 0.3μM, and of normal number and intensity at 1μM. Western blot analysis of SopB concentrations was consistent with these data (S2C Fig).

Accordingly, we tested whether foci initiated by wt SopB$^{N15}$ produced at 0.1 and 0.3μM arabinose could be expanded to visible size by spreading of SopB$^{N15/F}$::megfp produced at 1nM aTc, the optimal concentration for discrete focus formation on *sopC* (Fig 4A), and at 3nM aTc, for a moderate over-production (Fig 4B) to allow for the possibility that spreading in this heterologous system is less efficient. The results (Fig 4D) showed no focus formation with any combination of SopB$^{N15}$ and SopB$^{N15/F}$::megfp concentrations.

A variant of this approach, instigation of spreading of a ParB protein unable to bind specifically to an available centromere, is to use a mutant of a natural protein that lacks centromere-specific binding activity but is still capable of the non-specific DNA binding needed for spreading. The R219A derivative of SopB used in the experiment of Fig 2 is such a mutant. We essentially repeated this experiment using SopB$^F$ without the large peptides to which it had been fused to enable specific binding (Fig 5). After determining the inducer concentrations suitable for producing low levels of SopB (0.1μM arabinose) and SopB$^{F.R219A}$::megfp (1nM aTc), using

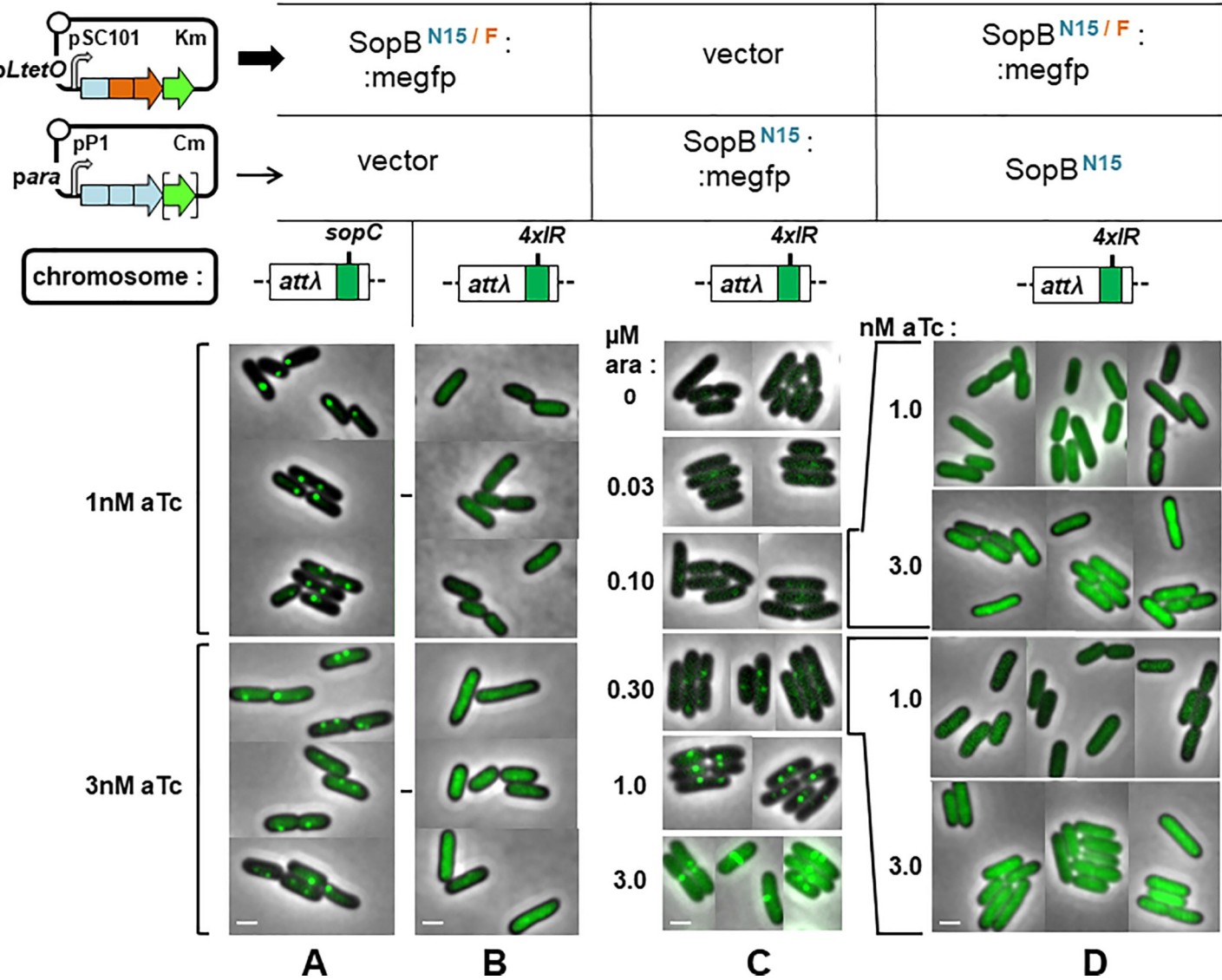

**Fig 4. Test of SopB spreading in the absence of a cognate centromere.** Cells of strains D195 (*sopC*) / pNR195 & pNR129, D183 (*4xIR*) / pNR195 & pNR129, D183 pNR197 & pZS21, and D183 / pNR189 & pNR129 grown exponentially in MGlyC with inducers for at least 10 generations, then viewed by fluorescence microscopy. Specificity of SopB$^{N15/F}$ binding shown by production at levels optimal (1nM aTc) or above (3nM) for visualization of complexes with **A.** sopC. **B.** 4xIR. **C.** Determination of minimal levels of arabinose-induced N15 SopB production needed to visualize complexes with 4xIR. **D**. Distribution of SopB$^{N15/F}$ in the presence of specific N15 SopB-centromere complexes: minimal specific complexes formed at 0.1 and 0.3 μM arabinose, as shown in fluorescent form in C, are tested for initiation of fluorescent complexes containing SopB$^{N15/F}$ produced at 1 and 3 nM anhydrotetracycline. Bar shows 1 μm.

silencing assays (S4B and S4C Fig; Fig 5A), we examined the ability of the former to initiate focus formation by the latter. No foci were seen.

These results provide no support for the proposal that interaction of SopB N-terminal domains alone can generate the spreading needed to assemble a functional partition complex.

## Discussion

In interpreting the results of our attempts to prime complex expansion from non-centromere DNA sites as the dependence of ParB protein spreading on cognate centromere binding, we

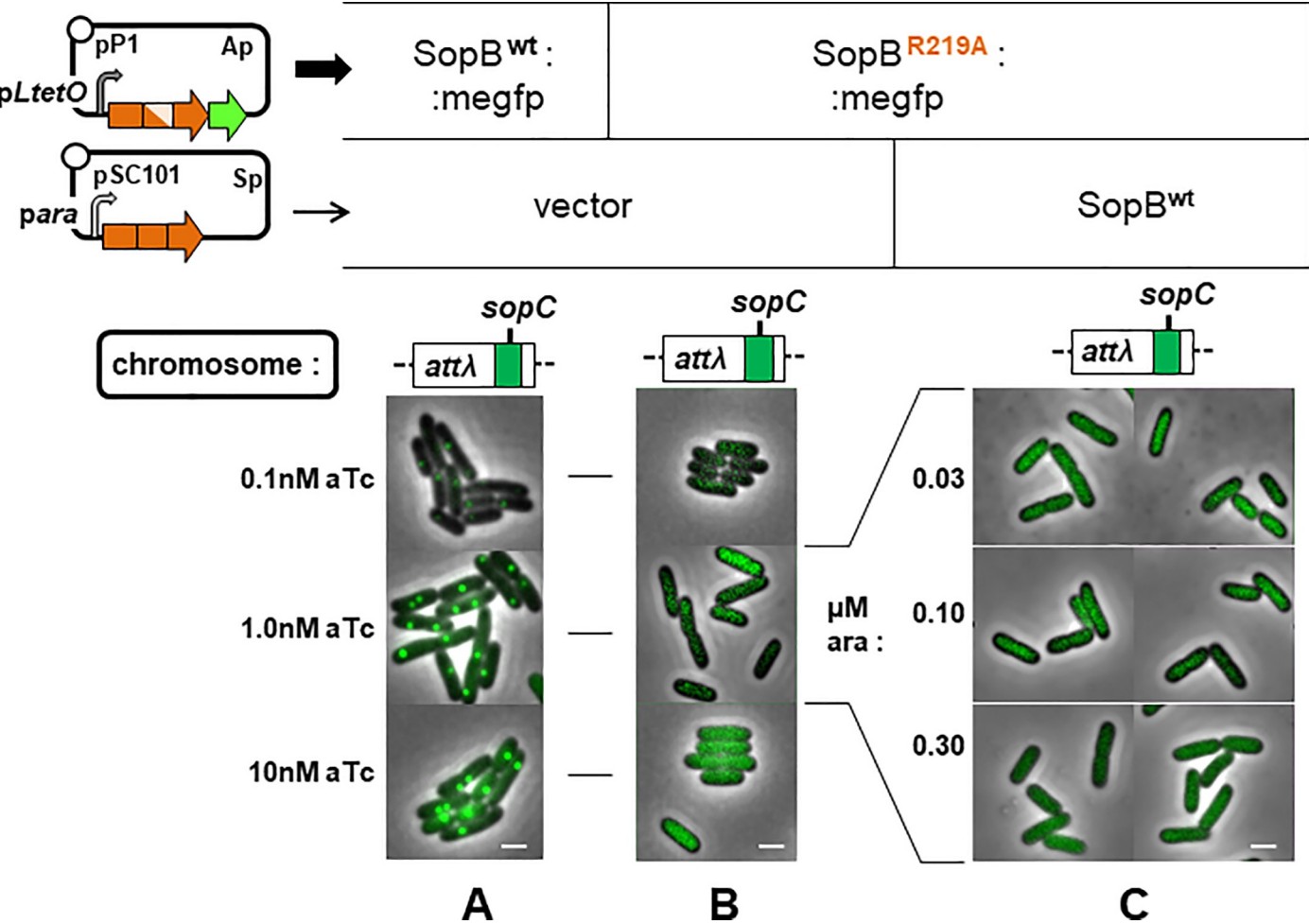

**Fig 5. Test of spreading by SopB unable to bind specifically to its centromere.** Cells of strain D195 carrying pDAG525 & pGB2, pNR198 & pGB2, and pNR198 & pDAG607 were grown and viewed as in Fig 4. Focus formation at different levels of **A.** wt SopB, **B.** R219A mutant SopB, **C.** wt and mutant SopBs. Bar shows 1 µm.

recognise two kinds of restriction. One is that this inference is based on the absence of focus formation rather than on a positive demonstration: we cannot rule out the possibility that use of other ParB proteins or alternative experimental approaches might reveal centromere-independent spreading.

The other is imposed by difficulties with the experimental material used here. Although Tale fusion protein binding appeared durable enough to enable ParB accumulation (S3A Fig, line 3), dCas9 fusions might have been too fleeting to trigger nucleation despite the effectiveness of the equivalent wt fusion. In addition, the possibility that the bulky Cas9 and Tale peptides to which ParBs were fused prevented acquisition of spreading competence seemed strong for several constructions (S3A Fig). It was therefore important to test centromere-independent spreading without them. The use of the hybrid SopB and the SopB mutant lacking specific binding activity served this purpose. That these proteins also failed to spread when primed by core complexes whose SopB proteins shared their N-terminal domains reinforces the original interpretation. It is still possible that the R219A mutant residue or the F component of the N15/F hybrid might interfere with the conformational transitions proposed to allow Nter domain interactions. This objection could in principle be met by modifying the hybrid SopB experiment to include the missing centromere, *sopC*, and observing whether this enabled

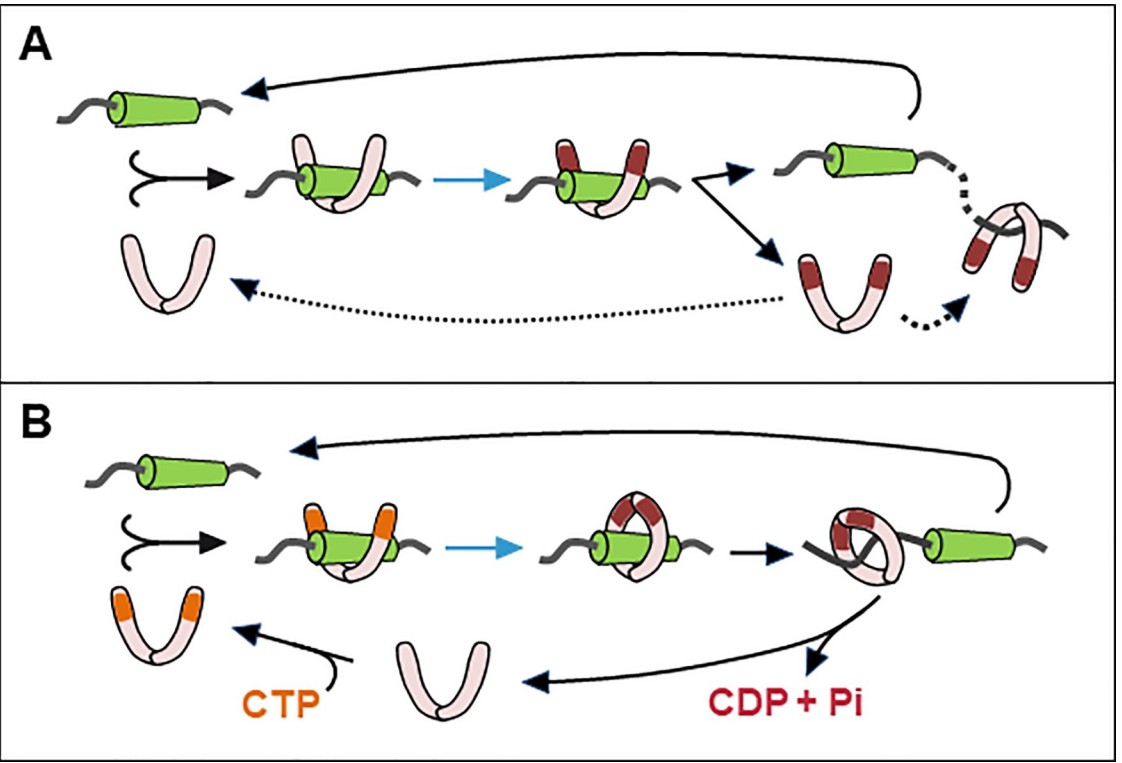

**Fig 6. Model of ParB Nter activation. A.** Proposed mechanism Biii from Fig 1; alternative fates of activated ParB after dissociation from *parS* are included. Green rod—*parS*; pink sausage—ParB monomer; maroon patch—activated Nter; blue arrow—supposed conformational switch to spreading competence. **B.** Spreading activated by CTP and *parS* binding, from Fig 4D in Soh et al (2019). For clarity the mechanism is shown as a series of steps, CTP binding (orange patches)—*parS* binding—ring closure, recognizing that the molecular mechanism may be more complex. CTP hydrolysis is assumed to cause dissociation of ParB from DNA and to allow recycling.

focus formation by the SopB$^{N15/F}$::megfp protein. However, the relaxed specificity of the N15 protein that enables it to bind functionally not only to its own centromere but also to that of F would complicate interpretation and prevent drawing a more definite conclusion.

If we accept, provisionally, that centromeres promote SopB spreading by specific activation rather than by simply focussing accretion of further dimers, we can examine the validity of the alternative bases of ParB activation summarized in Fig 1. ParB activation mechanisms based on spontaneous conversion or ParB-ParB interaction alone (options A) would be essentially eliminated. Option Aii could still contribute as a part of mechanism Bi, in which the centromere indirectly sequesters most of the diffusible ParB in a single conformer population through selective attraction of spontaneously arising active dimers. The intrinsically disordered nature of ParB Nter domains implies that they exist as a number of metastable, interchangeable conformers of which only one or a minority can spread, as underlined by the recent study of Hyde *et al.* [38], lending plausibility to mechanism Bi. However, if this proposal held, the SopB$^{N15/F}$ protein should assemble visible complexes with bound N15 SopB as readily as it does in the presence of its own centromere (Fig 4A), which it clearly did not (Fig 4D). The same objection can be raised to mechanism Bii, in which a dimer activated by centromere binding and residing there interacts with nascent dimers to induce their corresponding activation. This process, if applicable, also should have worked in the case of the N15/F protein. Notably, while mechanisms Bi and Bii both involve the centromere, the central selection or activation event occurs at the ParB protein interaction level.

Only mechanism Biii, successive and hence frequent binding and release of ParB dimers, depends solely on direct activation of ParB by the centromere, and it alone of those proposed appears to be consistent with our results. It does, however, raise the question of its compatibility with the demonstrated properties of ParB proteins, exemplified by SopB. The cohesiveness of the partition complex implies that the large majority of the cell's SopB dimers (~95%, [29]) are activated for spreading. Data from surface plasmon resonance and *in vivo* footprinting analyses [35,60] indicate a lengthy residence time, well in excess of a turnover time of about one minute per dimer per binding site, which we calculate (from 850 dimers in an average cell growing at two generations per hour [60]) would be needed to generate this activated majority, and of the 50-second half-life of Anchor3 complexes measured in human cells [4]. To reconcile our *in vivo* observations with the SPR and footprinting data it appeared necessary to posit a cellular element needed for rapid release of activated dimers that is absent from the *in vitro* assays. And indeed, two groups have very recently identified such an element—cytidine triphosphate (CTP; [36,37]). These authors discerned conserved motifs in the Nter region of several ParBs, including SopB, that enable binding of CTP. The binding was strongly stimulated by centromere DNA. In the case of the *B.subtilis* protein, binding to *parS* and to CTP induced interaction between Nter domains to form, as the major product, a dimer ring. Stimulation of ring formation by *parS* at sub-stoichiometric levels suggested that the rings vacate their binding site rapidly to slide along adjacent DNA, i.e. to spread; the process is schematized in Fig 6B, together with our option Biii (Fig 6A) to illustrate its correspondence. If future work shows CTP-SopB-*sopC* to behave in this way, the discrepancy between our focus-formation and *in vitro* binding results would disappear.

Given that activation of spreading ability depends on direct contact with the centromere, efforts to bring about ANCHOR visualization without it would now appear futile barring technical innovation. On the other hand, it might be possible to create mutant ParBs predisposed to adopt a spreading-competent conformation independently of their centromeres. The energy barrier to such conformers may well be low; Soh *et al.* [36] observed CTP to stimulate some formation of *B.subtilis* ParB dimer rings in the absence of *parS*, presumably from a subset of suitable conformers normally promoted by *parS* binding. A search for suitable mutants is clearly a priority.

## Supporting information

**S1 Fig. *E.coli* strains, genotype and derivation.** All transductions except the first involved cotransduction with a selective gene subsequently removed by FLP-mediated excision. Superscripts denote previously published strains (see Table 1). Lysogenization by λRS phages to integrate promoter-*lacZ* fusions has been described [40]. p*cry* denotes a cryptic promoter. (PPTX)

**S2 Fig. Western blot estimation of fusion proteins. A.** Cells of strain DLT1215 and of derivatives carrying pDAG114 (wt mini-F; [59]), pCAT05 (*sopB*^R219A^::d*cas9*) and pCAT15 (*sopB*^+^d-*cas9*), from cultures growing exponentially in MGC medium, Quantities of the R219A and wt SopB::dCas9 fusions (shaded arrowheads) relative to that of mini-F (~ 800 dimers/cell; clear arrowheads) were, respectively, 0.37 and 0.78 (left panel), and 1.1 and 0.20 (right panel), estimated using Image Lab (Biorad). Efficiency of SopB::dCas9 fusion protein transfer varied from one experiment to another: we show the results of two Western analyses of the same samples, which used MOPS buffer (left panel; irrelevant lanes between the third and fourth have been excised) and Tris-glycine (right panel) for electrophoresis and transfer. **B.** Exponential-phase cells of yeast strain W303 and derivatives harbouring plasmids that carry *megfp*::*parB*::*tal* fusions. The ladder shows prestained protein MW standards. Arrowed bands are, from left

to right, those of fusions ParB$^{Ralstonia\ sp.}$::Tale$^{U3M}$ (formula MW 159kD), ParB$^{S.pneumoniae}$::Tale$^{oL1\lambda}$ (153kD), ParB$^{S.pneumoniae}$::Tale$^{U3M}$ (149kD), Or3::Tale$^{U3M}$ (157kD); the band (pVR252) is Or3::mEgfp (66kD) without a Tale. The first three fusions are produced from the pHIS3 promoter, the last two from the stronger pTEF promoter. **C.** As in A, except that instead of the SopB::dCas9 fusions, wt SopB$^{N15}$ in strain D183/pNR189 and wt SopB$^{F}$ in D195/pDAG607 were analyzed. Arrowhead points to SopB from mini-F. Wedges represent graded arabinose inducer concentrations—0.1, 0.3, 1.0 nM and 0.03, 0.1, 0.3 nM respectively–used in the focus formation experiment of Fig 4.
(PPTX)

**S3 Fig. Tests of focus formation by ParB and Tale fusion constructions in *S.cerevisiae*. A.** Examples: 1 –mEgfp::Or3 forms one focus per cell in the presence (left panel) but not in the absence (right panel) of an integrated Anch3 site; 2 –fusion of tal.U3M to mEgfp::Or3 results in addition of foci with or without the Tale target site, implying susceptibility of tripartite protein to aggregation; 3 –mEgfp::ParB::Tale.oLλ protein forms one strong focus per cell by simple, FROS-like binding in cells with a target site array (left), not in cells without (right), implying normal binding properties of tripartite protein; 4 –exchange of ParB unit in tripartite protein above results in occasional, target site-independent foci, implying functional incompatibility of the new ParB; 5 –exchange of Tale unit for tal.U3M does not result in new focus; 6 –tripartite protein produced from moderate-strength promoter frequently forms a single focus per cell (though independently of target site), but 7—when produced from a stronger promoter forms several foci both outside and inside nucleus, impying aggregation rather than partition complex assembly. **B.** Other configurations used in attempts to observe partition complex foci; none gave rise to single foci in cells carrying the Tale target sequence.
(PPTX)

**S4 Fig. Estimation of partition complex expansion using promoter silencing. A.** Derivatives of strain D183 (4xIR—p*cry*::*lacZ*) carrying pNR189 (*sopB*$^{N15}$; clear circles) or pNR197 (*sopB*$^{N15}$::*megfp*; green circles) were grown exponentially for at least 10 generations in MGlyC with various concentrations of arabinose inducer, and culture samples assayed for β-galactosidase activity. Specific activity in the absence of arabinose was 17 Miller units. **B.** Strain D195 (*sopC*—p*ldc*::*lacZ*) carrying pDAG607 (*sopB*$^{F}$) and pNR198 (*sopB*$^{F.R219A}$) was grown and assayed as in A. Specific activity without arabinose was 525 MU. **C:** D195 carrying pDAG525 (*sopB*$^{F}$::*megfp*) was grown and assayed as in A with various concentrations of anhydrotetracycline. Specific activity without arabinose was 436 MU.
(PPTX)

**S1 Table. Interaction of target sequences with Cas9 guide RNAs.** Strains carrying chromosomal *xylE* and *sopC* sequences on the chromosome and genes for the corresponding sgRNAs on an expression vector were transformed with plasmids from which production of the SopB::dCas9 fusion or the equivalent active Cas9 fusion protein could be induced. Viability of transformants on agar medium was scored (see Materials & methods).
(DOCX)

## Acknowledgments

We thank Céline Benoit for strain and plasmid constructions and for performing killing assays, Ludmila Recoules for her expertise in Western blotting, Franck Gallardo for help in applying ANCHOR3 to yeast, Virginie Ramaillon for initial TALE concstructions, Bing Yang and Addgene for supplying plasmids and Dhruba Chattoraj for providing *E.coli* strains.

## Author Contributions

**Conceptualization:** Jean-Yves Bouet, Kerstin Bystricky, David Lane.

**Funding acquisition:** Jean-Yves Bouet, Kerstin Bystricky.

**Investigation:** Sylvain Audibert, Nicolas Tanguy-le-Gac, Jérôme Rech, David Lane.

**Project administration:** Jean-Yves Bouet, Kerstin Bystricky, David Lane.

**Resources:** Catherine Turlan.

**Supervision:** Jean-Yves Bouet, Kerstin Bystricky, David Lane.

**Validation:** Jérôme Rech, Jean-Yves Bouet, David Lane.

**Visualization:** Sylvain Audibert, Nicolas Tanguy-le-Gac, Jérôme Rech, David Lane.

**Writing – original draft:** David Lane.

**Writing – review & editing:** Sylvain Audibert, Nicolas Tanguy-le-Gac, Jean-Yves Bouet, Kerstin Bystricky, David Lane.

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
