## [Decision Letter · Decision Letter 0]

24 Dec 2019

PONE-D-19-33138

Role of centromere sites in activation of ParB proteins for partition complex assembly

PLOS ONE

Dear Lane,

Thank you for submitting your manuscript to PLOS ONE. After careful consideration, we feel that it has merit but does not fully meet PLOS ONE’s publication criteria as it currently stands. Therefore, we invite you to submit a revised version of the manuscript that addresses the points raised during the review process.

Specifically, please include additional characterization of your system to substantiate your claim that the failure of chimeric ParB to produce foci was indeed due to its inability to bind ParS and not due to other uninteresting problems with construction of chimeric proteins. Reviewer 1 suggested a number of important controls to this end. I would also recommend a positive control, when you join the N- and C-terminal domains of ParB via the same linker that you use in other chimera and observe ParS-dependent formation of the foci. 

There is also a logical gap between your observations and the conclusion as pointed out by the Reviewer 2. It is really impossible to draw such a strong conclusion as you propose from a negative result. Please reword the title of your manuscript to better reflect your observations and the revise the text of the manuscript accordingly. 

We would appreciate receiving your revised manuscript by Feb 07 2020 11:59PM. To enhance the reproducibility of your results, we recommend that if applicable you deposit your laboratory protocols in protocols.io, where a protocol can be assigned its own identifier (DOI) such that it can be cited independently in the future. For instructions see: http://journals.plos.org/plosone/s/submission-guidelines#loc-laboratory-protocols

We look forward to receiving your revised manuscript.

Kind regards,

Valentin V Rybenkov

Academic Editor

PLOS ONE

Journal Requirements:

2. We note you have included a table to which you do not refer in the text of your manuscript. Please ensure that you refer to Table 4 in your text; if accepted, production will need this reference to link the reader to the Table.

Reviewers' comments:

Reviewer's Responses to Questions

**Comments to the Author**

1. Is the manuscript technically sound, and do the data support the conclusions?

Reviewer #1: Yes

Reviewer #2: No

2. Has the statistical analysis been performed appropriately and rigorously? 

Reviewer #1: N/A

Reviewer #2: N/A

3. Have the authors made all data underlying the findings in their manuscript fully available?

Reviewer #1: Yes

Reviewer #2: Yes

4. Is the manuscript presented in an intelligible fashion and written in standard English?

Reviewer #1: Yes

Reviewer #2: No

5. Review Comments to the Author

Reviewer #1: ParB proteins have the unique ability to accumulate in high copy numbers at a defined DNA recognition sequence. They are employed to visualize selected loci on a chromosome. This labeling approach requires integration of target sequences into the genome and expression of fluorescent ParB fusion proteins. The manuscript by Audibert et al. aims to simplify locus labeling by recruiting ParB to (more) freely selectable target sequences and to improve the understanding of the underlying mechanism. To do so, the presented work targets ParB proteins to alternative DNA sequences by fusion with unrelated DNA-binding proteins (dCas9 and Talen) in order to artificially nucleate local ParB enrichment.

While most of the results presented in this manuscript are negative, they are still well-worth reporting. Overall, the conclusion is that ParB proteins cannot easily be targeted to alternative DNA sequences, likely because nucleation of local ParB enrichment is tightly linked with target sequence recognition by ParB. The findings are largely consistent with the recently reported discovery of an enzymatic activity of ParB, which appears essential for local enrichment of ParB. The study is well designed, and the experiments are executed to a high standard. The experiments involving chimeric SopB proteins are particularly elegant.

The figures already include several controls that show that the experiments are working as intended. However, two (hopefully easy) control experiments should be added to make sure that the negative results are not caused by obvious experimental problems.

A) Can the expression levels of tagged and untagged ParB variants be quantified? Semi-quantitative Western blotting may be sufficient to show that all proteins are present in reasonable amounts.

B) All experiments rely on the specific binding of an engineered or mutant protein (dCas9 or TALEN fusion proteins and SopB R219A mutants) to a target sequence. The efficiency of the specific binding is however not directly tested for any of the presented approaches. Cell killing by Cas9 implies target sequence recognition. However, very transient or rare binding might be sufficient for cell killing, while ParB nucleation might require more stable binding. Can the localization of these proteins be demonstrated directly by ChIP-sequencing or ChIP-qPCR?

Local enrichment requires energy input. Simple uncoupling of CTP binding/hydrolysis from target sequence recognition (as alluded to at the end of the discussion) will accordingly disrupt ParB accumulation even at heterologous sites. Successful artificial targeting should require mechanistic linkage of CTP binding/hydrolysis with the (then alternative) DNA binding of the fusion partner (dCas9 or TALEN). The statement should thus be softened.

Fig. 1 (or a final figure) should include the recently proposed DNA spreading model.

A short introduction of the N15 prophage might be helpful for the less experienced reader.

Page 15, 2nd paragraphs: Please provide a list of all conditions tested for targeting (different ParBs, tripartite fusion…). Even without full documentation, the information may be useful to some readers.

Fig. 3: The ParB foci in the yeast images are quite faint. Please provide arrowheads or enhanced contrast.

Reviewer #2: The author attempts to observe ParB spreading by the interaction of its N-terminal domain alone, rather than ParB-ParS interactions. To test the possibility, two ParB family proteins in E.coli and yeast, SopB and Or3 were fused to non-parS DNA binding fluorescent proteins. To this end, they are used to distinguish ParB-parS binding from ParB-ParB interaction. The results provide no support for the proposal that interaction of SopB N-terminal domains alone can generate the spreading needed to assemble a functional partition complex. The author finally conclude ParB proteins need their cognate centromeres to become capable of spreading.

At current level, the hypothesis raised in this work was denied by the experiment result, and no new knowledge was created. The conclusion, parB proteins need their cognate centromeres to become capable of spreading, has been extensively approved in various species. This is not the conclusion from this work. On the other hand, the context structure, logical connection and details of the language of this manuscript need improving to reach the publish level.

6. PLOS authors have the option to publish the peer review history of their article (what does this mean?). If published, this will include your full peer review and any attached files.

Reviewer #1: No

Reviewer #2: No

---

## [Author Response · Author response to Decision Letter 0]

28 Feb 2020

Dear Dr Rybenkov

 We have considered the the reviewers' comments and your own concerning our paper "Role of centromere sites in activation of ParB proteins for partition complex assembly", PONE-D-19-33138. 

 Before detailing our responses I should point out that the original MS included two errors :

First, the version of Fig 1 submitted is not the one that corresponds to the Fig 1 legend or to the text ; the portions Bi and Biii were exchanged. This might have contributed to difficulties in following the logic of the paper. The version resubmitted has been corrected.

Second, the description of the construction of a series of plasmids, under "SopB::mVenus visualization" was far from complete. This has now been corrected, see p.7.

Other points :

Addition of two new supplementary figures and renumbering to preserve the correct order.

A number of minor corrections and modifications, not worth listing but readily seen in the marked up file.

The revisions necessitated addition of three more references (numbers 48, 58 and 60 in the new MS). We have also added a fourth (number 37, Osorio-Valeriano et al) because it represents the same major advance as the Soh et al paper already cited but had not come out at the time of our original submission.

Editor's comments :

1 - Specifically, please include additional characterization of your system to substantiate your claim that the failure of chimeric ParB to produce foci was indeed due to its inability to bind ParS and not due to other uninteresting problems with construction of chimeric proteins. 

Reply: By "inability to bind parS" we presume you mean "inability to spread in the absence of parS", since this is our main conclusion. We agree that difficulties in production and behaviour of the chimeric proteins could limit the certainty of this conclusion. We have added Western blot data (Fig S2), as also suggested by Reviewer 1, to demonstrate normal levels of chimeric protein production, and a sentence to point this out (top p.16). Although in several cases (Figs 4C, 5A, S3A, ...) the chimeric proteins are still capable of binding parS, whereas in others we cannot tell whether they do or not owing to apparent abnormal behaviour (your "uninteresting problems" presumably) of the large chimeras. This was acknowledged in the original MS on p.15 following Fig. 3 legend "... formation of aggregates rather than bona fide foci ... " and is evident in some of the examples we now provide in Fig S3. We have also underlined this phenomenon, and thus softened our conclusion, with a modified sentence (the third) of the Discussion. In fact it is exactly these limits to our conclusions of the large protein fusion experiments that lead to and justify the hybrid SopB protein experiments in which centromere binding is demonstrably not a problem and which buttress our conclusion that centromere binding is needed for focus formation.

2 - Reviewer 1 suggested a number of important controls to this end. I would also recommend a positive control, when you join the N- and C-terminal domains of ParB via the same linker that you use in other chimera and observe ParS-dependent formation of the foci.

Reply: It is not clear what new constructions are being suggested. Our general experience with joining ParB proteins to Gfp-type peptides, Nter or Cter (refs 3-5 in the MS), is that focus formation is achieved with a wide variety of linker peptides. If the order of the Nter and Cter domains of the N15-F hybrid is in question, we know (but saw no need to say) that only the hybrid used in the paper is functional, and that the equivalent F-N15 hybrid is not. If the ParB::Cas9 and ParB::Tale fusions are the point, we made no systematic attempt to test variations of linker peptide sequence or length, particularly as Tale::ParB fusions proved far more difficult to make and maintain (none appear in our Fig S3 summary) than ParB::Tale fusions, regardless of linker sequence. 

3 - There is also a logical gap between your observations and the conclusion as pointed out by the Reviewer 2. It is really impossible to draw such a strong conclusion as you propose from a negative result.

Reply: We do not agree that there is a logical gap; Reviewer 2 certainly does not explain it (see comment below). We propose certain possibilities (Fig 1), test them in the experiments subsequently described, and find that only one of the possibilities is not ruled out by the results. We do not claim that this is a strong conclusion, rather that it is the only one left standing, as must be the case in the absence of a positive demonstration. The very recent provision of such a demonstration (Soh et al, Osorio et al) reinforces our conclusion but we do not pretend that this alters its essentially provisional nature.

4 - Please reword the title of your manuscript to better reflect your observations and the revise the text of the manuscript accordingly. 

Reply: The title is entirely neutral. It does not say in advance what we conclude the role of centromere sites to be, or even that there necessarily is one. If one wishes to look behind its literal meaning, the title implies that centromere sites have already been proposed to have a role in ParB activation, as Reviewer 2 points out, and this study is aimed at defining that role. We do not understand why Reviewer 2 or you would object to this.

5 - 2. We note you have included a table to which you do not refer in the text of your manuscript. Please ensure that you refer to Table 4 in your text; if accepted, production will need this reference to link the reader to the Table.

Reply: We have now included reference to Table 4, under "Plasmid constructions" on p.6.

Reviewer #1: 

The figures already include several controls that show that the experiments are working as intended. However, two (hopefully easy) control experiments should be added to make sure that the negative results are not caused by obvious experimental problems.

A) Can the expression levels of tagged and untagged ParB variants be quantified? Semi-quantitative Western blotting may be sufficient to show that all proteins are present in reasonable amounts.

We have included Western blot data as Fig. S2. They show that for the three experimental situations examined the amounts of full-size ParB fusion protein are at least at the levels at which the corresponding wild type ParB proteins would form partition complexes. In the case of the SopB fusion proteins this conclusion is reached by comparison with wt SopB. In the case of the Gfp::ParB::Tale fusions it is relative to visible fluorescence levels; Fig S2B shows that the full-sized fusions are the only significant anti-Gfp species present.

B) All experiments rely on the specific binding of an engineered or mutant protein (dCas9 or TALEN fusion proteins and SopB R219A mutants) to a target sequence. The efficiency of the specific binding is however not directly tested for any of the presented approaches. Cell killing by Cas9 implies target sequence recognition. However, very transient or rare binding might be sufficient for cell killing, while ParB nucleation might require more stable binding. Can the localization of these proteins be demonstrated directly by ChIP-sequencing or ChIP-qPCR?

In vivo binding of dCas9 is generally rather stable (occupancy >30 mins E.coli, Jones et al Science 357, 1420-1424; >3hrs eucaryotes, Ma et al J. Cell Biol. 214, 529-537), though it is possible that at the sites used transient binding of a dCas9-parB chimera is insufficient to trigger nucleation. We have added a sentence to the start of the Discussion to cover this possibility. But whether durable or transient, binding did not enable nucleation, which was the aim of creating the chimeric proteins in the first place. 

In any case ChiP is not a method of choice for measuring efficiency of binding of a single protein. It can report the presence of proteins at a given DNA locus above background, as in its successful use to demonstrate ParB::GFP spreading in yeast (Saad) and SopB::mVenus in bacteria (Sanchez). Without amplification by spreading, variation in antibody specificity and PCR efficiency makes detection of even tight binding of a single molecule unreliable. Restriction site protection would be a possibility but given the unlikelihood of brief occupancy being the problem we did not consider the investment worthwhile.

Local enrichment requires energy input. Simple uncoupling of CTP binding/hydrolysis from target sequence recognition (as alluded to at the end of the discussion) will accordingly disrupt ParB accumulation even at heterologous sites. Successful artificial targeting should require mechanistic linkage of CTP binding/hydrolysis with the (then alternative) DNA binding of the fusion partner (dCas9 or TALEN). The statement should thus be softened.

The reviewer appears to propose something other than what we are suggesting at the end of the discussion. It is hard to see how the alternative binding partner could be made to transmit the appropriate signal to the fused ParB. In any case it is the parB we would mutate, in the hope that the low energy barrier to CTP-induced transition already suggested by Soh et al's data (Fig. 3C) could be further lowered to the point where spontaneous transition to active conformer state replaces parS-mediated stimulation often enough to enable a focus to become visible at the alternative binding site.

We considered inserting this suggestion into the final figure recommended by the reviewer (see next) but given that isolation of an appropriate ParB mutant is entirely speculative we decided such an addition premature. 

Fig. 1 (or a final figure) should include the recently proposed DNA spreading model.

The recent model is not fundamentally different from Biii, rather it is extended and far more concrete, so addition to Fig. 1 is not appropriate. We have followed the reviewer's suggestion to add a final figure, Fig. 6, that stresses the relationship between the two, hoping that this is what the reviewer had in mind. A few words added at the bottom of p.21 refer to it. 

A short introduction of the N15 prophage might be helpful for the less experienced reader.

We have added a sentence at the bottom of p.16 to fill this gap.

Page 15, 2nd paragraphs: Please provide a list of all conditions tested for targeting (different ParBs, tripartite fusion…). Even without full documentation, the information may be useful to some readers.

We have added a supplementary figure (Fig. S3) in two parts, one (A) with images illustrating particular points (parS binding, Tale target binding, interference by aggregates), the other (B) as a list of constructions tried and tested.

Fig. 3: The ParB foci in the yeast images are quite faint. Please provide arrowheads or enhanced contrast.

The problem here was in the conversion to .pdf images. The final on-line version should be much clearer.

Reviewer #2: 

At current level, the hypothesis raised in this work was denied by the experiment result, and no new knowledge was created. 

We concede, obviously, that the result was negative in the sense that no evidence was obtained for purely ParB-ParB interaction as a basis for spreading, but this is not the same as "no new knowledge". The essential message is "if you are thinking about trying centromere-independent spreading, think again, it has already been tried."

The conclusion, parB proteins need their cognate centromeres to become capable of spreading, has been extensively approved in various species. 

There is a distinction to be made here, as we explain in the Introduction (bottom, p.3) : does the need for parS in spreading consist of activating ParB to enable it, or of providing a focal point to concentrate ParB molecules which are activated by other means ? No previous study has explicitly attempted to discriminate between these two options. Several authors have suggested direct parS-mediated activation but this was opened to doubt by other observations (Surtees & Funnell; Hyde et al), as detailed in the Introduction. The reviewer goes with the common assumption but this is not the same as a demonstration. In fact it is only the very recent report by Soh et al that puts the issue beyond reasonable doubt. Our conclusion is consistent with it. 

This is not the conclusion from this work. 

If we understand the reviewer, this is exactly the conclusion of this work. It is just that we have also sought to examine other possible mechanisms.

On the other hand, the context structure, logical connection and details of the language of this manuscript need improving to reach the publish level.

The reviewer does not explain what problems he sees with "the context structure" and "logical connection", so there seems nothing to reply to. Likewise, what the reviewer means by improvement in "the language of this manuscript" is not explained.

---

## [Decision Letter · Decision Letter 1]

10 Apr 2020

PONE-D-19-33138R1

Role of centromere sites in activation of ParB proteins for partition complex assembly

PLOS ONE

Dear Lane,

Thank you for submitting your manuscript to PLOS ONE. After careful consideration, we feel that it has merit but does not fully meet PLOS ONE’s publication criteria as it currently stands. Therefore, we invite you to submit a revised version of the manuscript that addresses the points raised during the review process.

Specifically, I ask you again to re-evaluate your conclusions, which at times appear to reach further than warranted by the data. This is a classic case of when a failure to prove a phenomenon does not prove its absence. I agree with the reviewer that your findings carry a constructive message. However, by overinterpreting them, you negate their impact. Please note that one of PLOS ONE publication criteria states: "Conclusions are presented in an appropriate fashion and are supported by the data". In my view, this means that conclusions must be clearly separated from inferences. 

Please also note other reveiewer's suggestions, which would undoubtfully improve the quality of your manuscript.

We would appreciate receiving your revised manuscript by May 25 2020 11:59PM. To enhance the reproducibility of your results, we recommend that if applicable you deposit your laboratory protocols in protocols.io, where a protocol can be assigned its own identifier (DOI) such that it can be cited independently in the future. For instructions see: http://journals.plos.org/plosone/s/submission-guidelines#loc-laboratory-protocols

We look forward to receiving your revised manuscript.

Kind regards,

Valentin V Rybenkov

Academic Editor

PLOS ONE

Reviewers' comments:

Reviewer's Responses to Questions

**Comments to the Author**

1. If the authors have adequately addressed your comments raised in a previous round of review and you feel that this manuscript is now acceptable for publication, you may indicate that here to bypass the “Comments to the Author” section, enter your conflict of interest statement in the “Confidential to Editor” section, and submit your "Accept" recommendation.

Reviewer #1: All comments have been addressed

2. Is the manuscript technically sound, and do the data support the conclusions?

Reviewer #1: Yes

3. Has the statistical analysis been performed appropriately and rigorously? 

Reviewer #1: N/A

4. Have the authors made all data underlying the findings in their manuscript fully available?

Reviewer #1: Yes

5. Is the manuscript presented in an intelligible fashion and written in standard English?

Reviewer #1: Yes

6. Review Comments to the Author

Reviewer #1: The authors have made several modifications to the manuscript text and added control experiments in response to the reviewers’ and editor’s comments. They have seriously considered the comments and have made appropriate changes or alternatively provided explanations in the written response to the comments.

While the outcomes remain negative, the manuscript with its well-executed experiments will be a valuable addition to the literature, at the least instructing other researchers when attempting similar experiments in the future. Altogether, this reviewer recommends publication of the work in its current form. The authors may however consider the points given below prior to publication.

Minor points:

Following the comments from the other reviewer and the editor, the title of the manuscript could be slightly toned down to ‘Addressing the role of centromeric sites in activation of ParB proteins for partition complex assembly’.

Throughout the manuscript the involvement of ParB oligomerization in ParB spreading is implicated. However, the conclusions of this work are largely independent thereof. Thus, the authors may want to consider making their work stand regardless of ParB oligomerization by eliminating such statements.

The figure legends are rather tough to read, mainly due to the listing of many genotypes. Information on genotypes could be removed from the legends altogether. The information is largely provided in the figure panels. Only strain names and plasmid names could be included in the figure legends or alternatively the use of plasmids and strains including genotypes for each figure could be listed in a separate table.

The figure legend to Figure S4 is particularly long. Please consider moving information on the Western blot protocol to the methods section.

Panel designators are not consistently noted throughout the figure legends. A : in Figure 4. A- and (A) in Figure 3. A – in Figure 1. Please revise.

7. PLOS authors have the option to publish the peer review history of their article (what does this mean?). If published, this will include your full peer review and any attached files.

Reviewer #1: No

---

## [Author Response · Author response to Decision Letter 1]

14 Apr 2020

Editor:

Specifically, I ask you again to re-evaluate your conclusions, which at times appear to reach further than warranted by the data. This is a classic case of when a failure to prove a phenomenon does not prove its absence. I agree with the reviewer that your findings carry a constructive message. However, by overinterpreting them, you negate their impact. Please note that one of PLOS ONE publication criteria states: "Conclusions are presented in an appropriate fashion and are supported by the data". In my view, this means that conclusions must be clearly separated from inferences. 

Reply: We can only agree with the principle at issue here, though apparently you consider our attempts to deal with it in our first revision (between [ ] below) inadequate. It would have been helpful if you could have provided a clearer indication of what you mean by "overinterpretation", using one or two specific examples from the text for instance. We can only suppose that you see overinterpretation in either or both of two ways : (i) any attempt to discuss the results beyond the simple observation of lack of focus formation by hybrid ParB proteins is unjustified ; (ii) we have been too assertive in summary or linking sentences – "concluding" rather than "inferring", for example.

If your objection is the former, we would reply as follows. The major conclusion – that ParB proteins do not spread in the absence of their centromeres – is clearly provisional, being based in this paper only on the proteins tested and our experimental approach (one can hardly be expected to test every conceivable configuration). It does however correspond to the general assumption in the field (as seen by the attitude of reviewer 2). But once it is provisionally accepted the issue becomes what the centromere is needed for, which is why we discuss (initially in the 2nd, now the 3rd paragraph of the Discussion) the options outlined in Fig 1 and how our results bear on them, leading ineluctably to centromere activation of ParB spreading. If this is not suitable for discussion then not only does the Discussion lose most of its interest but we would have fallen far short of "Addressing" (as the reviewer recommends we modify the title) the topic. It would also render superfluous both the fuller perspective provided by Fig 1and the Fig 6 recommended by the reviewer.

If your objection is our manner of expression then you may have a point : readers might well get the impression of stronger finality in our statements than we intended to convey. On this assumption, we have modified the text at various points, including the Abstract, as shown in the marked up copy. In particular, we have modified the start of the Discussion to squarely explain the limits of our interpretation, and on p.20 emphasized the provisional nature of the deduced centromere role.

[1 - Specifically, please include additional characterization of your system to substantiate your claim that the failure of chimeric ParB to produce foci was indeed due to its inability to bind ParS and not due to other uninteresting problems with construction of chimeric proteins. 

Reply: By "inability to bind parS" we presume you mean "inability to spread in the absence of parS", since this is our main conclusion. We agree that difficulties in production and behaviour of the chimeric proteins could limit the certainty of this conclusion. We have added Western blot data (Fig S2), as also suggested by Reviewer 1, to demonstrate normal levels of chimeric protein production, and a sentence to point this out (top p.16). Although in several cases (Figs 4C, 5A, S3A, ...) the chimeric proteins are still capable of binding parS, whereas in others we cannot tell whether they do or not owing to apparent abnormal behaviour (your "uninteresting problems" presumably) of the large chimeras. This was acknowledged in the original MS on p.15 following Fig. 3 legend "... formation of aggregates rather than bona fide foci ... " and is evident in some of the examples we now provide in Fig S3. We have also underlined this phenomenon, and thus softened our conclusion, with a modified sentence (the third) of the Discussion. In fact it is exactly these limits to our conclusions of the large protein fusion experiments that lead to and justify the hybrid SopB protein experiments in which centromere binding is demonstrably not a problem and which buttress our conclusion that centromere binding is needed for focus formation.]

Reviewer #1: 

The authors have made several modifications to the manuscript text and added control experiments in response to the reviewers’ and editor’s comments. They have seriously considered the comments and have made appropriate changes or alternatively provided explanations in the written response to the comments.

While the outcomes remain negative, the manuscript with its well-executed experiments will be a valuable addition to the literature, at the least instructing other researchers when attempting similar experiments in the future. Altogether, this reviewer recommends publication of the work in its current form. The authors may however consider the points given below prior to publication.

Minor points:

Following the comments from the other reviewer and the editor, the title of the manuscript could be slightly toned down to ‘Addressing the role of centromeric sites in activation of ParB proteins for partition complex assembly’.

Reply: We have adopted the title suggested.

Throughout the manuscript the involvement of ParB oligomerization in ParB spreading is implicated. However, the conclusions of this work are largely independent thereof. Thus, the authors may want to consider making their work stand regardless of ParB oligomerization by eliminating such statements.

Reply: Oligomerization via N-terminal domains of ParB dimers has been a regular if seldom demonstrated part of ParB research for 20 years, so we think it is appropriate to refer to it in the Introduction. But as the the reviewer points out, our results do not bear directly on oligomerization itself . And the recent model of Soh et al scarcely involves it. So from Results on we have referred simply to "N-terminal domain".

The figure legends are rather tough to read, mainly due to the listing of many genotypes. Information on genotypes could be removed from the legends altogether. The information is largely provided in the figure panels. Only strain names and plasmid names could be included in the figure legends or alternatively the use of plasmids and strains including genotypes for each figure could be listed in a separate table.

Reply: We have removed plasmid genotypes from the legends to Figs 2, 4 and 5, leaving only strain and plasmid names.

The figure legend to Figure S4 is particularly long. Please consider moving information on the Western blot protocol to the methods section.

Reply: The reviewer means Fig S2. As suggested, the purely protocol portions have been transposed to a new Western blotting section in Materials & Methods.

Panel designators are not consistently noted throughout the figure legends. A : in Figure 4. A- and (A) in Figure 3. A – in Figure 1. Please revise.

Reply: Designations have been corrected for all legends, including those for Supplementary figures to the format A. .

---

## [Editor Report · Decision Letter 2]

16 Apr 2020

Addressing the role of centromere sites in activation of ParB proteins for partition complex assembly

PONE-D-19-33138R2

Dear Dr. Lane,

We are pleased to inform you that your manuscript has been judged scientifically suitable for publication and will be formally accepted for publication once it complies with all outstanding technical requirements.

With kind regards,

Valentin V Rybenkov

Academic Editor

PLOS ONE
---

## [Editor Report · Acceptance letter]

20 Apr 2020

PONE-D-19-33138R2 

Addressing the role of centromere sites in activation of ParB proteins for partition complex assembly 

Dear Dr. Lane:

I am pleased to inform you that your manuscript has been deemed suitable for publication in PLOS ONE. Congratulations! Your manuscript is now with our production department. 

With kind regards,

on behalf of

Dr. Valentin V Rybenkov 

Academic Editor

PLOS ONE